# Seeing Text in the Dark: Algorithm and Benchmark

## ABSTRACT

Localizing text in low-light environments is challenging due to visual degradations. Although a straightforward solution involves a two-stage pipeline with low-light image enhancement (LLE) as the initial step followed by detection, LLE is primarily designed for human vision rather than machine vision and can accumulate errors. In this work, we propose an efficient and effective single-stage approach for localizing text in the dark that circumvents the need for LLE. We introduce a constrained learning module as an auxiliary mechanism during the training stage of the text detector. This module is designed to guide the text detector in preserving textual spatial features amidst feature map resizing, thus minimizing the loss of spatial information in texts under low-light visual degradations. Specifically, we incorporate spatial reconstruction and spatial semantic constraints within this module to ensure the text detector acquires essential positional and contextual range knowledge. Our approach enhances the original text detector's ability to identify text's local topological features using a dynamic snake feature pyramid network and adopts a bottom-up contour shaping strategy with a novel rectangular accumulation technique for accurate delineation of streamlined text features. In addition, we present a comprehensive low-light dataset for arbitrary-shaped text, encompassing diverse scenes and languages. Notably, our method achieves state-of-the-art results on this low-light dataset and exhibits comparable performance on standard normal light datasets. The code and dataset will be released.

## CCS CONCEPTS

• **Computing methodologies** → **Computational photography**.

## KEYWORDS

Constrained Learning, Low-Light Text Localization

## 1 INTRODUCTION

Scene text detection plays a crucial role in multimedia understanding, laying the groundwork for tasks such as text recognition, information extraction, and scene understanding. Approaches to arbitrary scene text detection can usually be divided into two main categories. Generally, top-down methods [13, 22, 23, 25, 27, 33, 40] primarily focus on either directly or incrementally delineating the entire contour and area of the text. In contrast, typical bottom-up [10, 16, 21, 30, 31] approaches first identify individual text components and then assemble them together. Both top-down and

Permission to make digital or hard copies of all or part of this work for personal or classroom use is granted without fee provided that copies are not made or distributed for profit or commercial advantage and that copies bear this notice and the full citation on the first page. Copyrights for components of this work owned by others than the author(s) must be honored. Abstracting with credit is permitted. To copy otherwise, or republish, to post on servers or to redistribute to lists, requires prior specific permission and/or a fee. Request permissions from permissions@acm.org.

ACM MM, 2024, Melbourne, Australia

© 2024 Copyright held by the owner/author(s). Publication rights licensed to ACM.
ACM ISBN 978-x-xxxx-xxxx-x/YY/MM
https://doi.org/10.1145/nnnnnnn.nnnnnnn

bottom-up methods, each with their unique strengths, have undergone rapid development and achieved remarkable results in the field of arbitrary shape text detection. Although significant progress has been made in text detection, detecting arbitrary shape text in low light conditions remains a significant challenge. Insufficient lighting leads to visual degradations such as blurred details, reduced brightness and contrast, and distorted color representation, making it difficult for both humans and text detectors to locate text. Figure 1 illustrates some examples of the text in low-light scenes.

A straightforward two-stage approach to solving low-light text detection involves enhancing low-light text images using existing LLE methods such as [17, 38, 39], followed by applying text detectors to the enhanced images. However, mainstream LLE methods, tailored for better visual effect for human vision, often overlook the need of downstream detection tasks. This can lead to the degradation of text's inherent features through brightness or color adjustments, often resulting in detection failures, as shown in Figure 1. Furthermore, the lack of dedicated datasets for arbitrary shape text detection in low-light conditions has significantly hampered the validation of methods in real-world scenarios. This has led to an over-reliance on synthetic data, exacerbating a notable domain gap in low-light arbitrary shape text detection research.

Consequently, we step outside the "enhance-first, detect-later" framework and propose a one-stage solution without any pre-processing enhancement. To address the issue of spatial information loss in normal-light text detectors caused by degradation factors such as low illumination and low contrast, we design a spatial-constrained learning module during the training process. This module, guided by two constraints, i.e., Spatial Reconstruction Constraint and Spatial Semantic Constraint, effectively preserves and pinpoints both the positional and contextual range information of text, which are often at risk of being lost during the spatial resizing of feature maps.

We further enhance the text detector by focusing on the intrinsic characteristics of text and adapting text shaping methods to low-light scenarios. Unlike general objects, text possesses a unique topological distribution and streamline structure. We incorporate Dynamic Snake Convolution [19] (DSC), renowned for its prowess in preserving tubular topological features, alongside conventional convolutions to capture the intrinsic characteristics of text in parallel. Additionally, we design a self-attention gate to control the proportions of convolutions and construct a novel feature pyramid network [9] (FPN) structure, thereby significantly enriching the representation of local topological features of text through improved fusion steps.

Regarding text's streamline characteristics, the top-down text detection approaches are somewhat less suitable under low-light conditions due to limited receptive field and difficulties in global information acquisition. Therefore, we adopt a more flexible bottom-up modeling strategy, employing an innovative rectangular accumulation approach for text contour modeling. This strategy enables

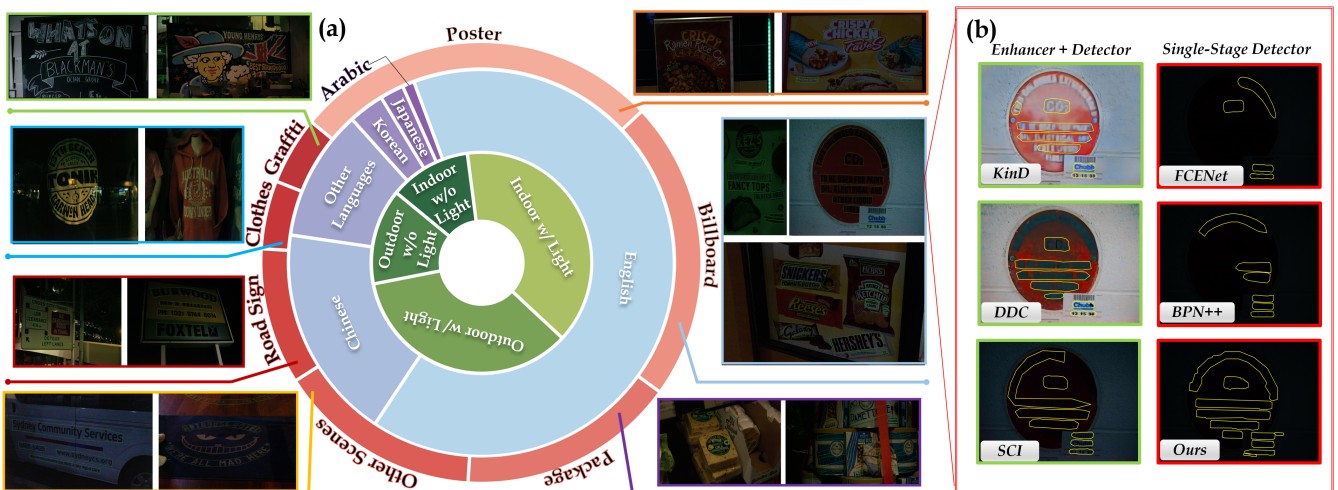

Figure 1: (a) The visual statistics and examples of low-light text images of different scenes, languages, and lighting conditions in the newly curated LATeD dataset. All images are enhanced for clearer vision. (b) Single-stage text detectors originally designed for normal lighting conditions struggle with low-light images. Even with low-light enhancement and fine-tuned with [35] following a two-stage step, the results remain unsatisfactory. This is because the enhancer, aimed at overall improvement of visibility, may inadvertently compromise text features.

the creation of a streamlined and flexible representation of text contours, alleviating the issue of limited receptive fields.

The major contributions of this paper are fourfold:

1) For the first time, we propose a one-stage pipeline for low-light arbitrary-shaped text detection that effectively utilizes spatial constraint to adeptly guide the training process.

2) We devise a novel method concentrating on the extraction of topological distribution features and the modeling of streamline characteristics to shape low-light text contours effectively.

3) We curate the first low-light arbitrary-shape text dataset (LATeD) featuring 13,923 multilingual and arbitrary shape texts across diverse low-light scenes, as shown in Figure 1, effectively bridging the existing domain gap.

4) Our method, employing a constrained learning strategy and capturing intrinsic text features, attains state-of-the-art results on low-light text detection dataset without image enhancement modules and also excels over SOTAs on similarly well-lit datasets.

## 2 RELATED WORKS

**Arbitrary Shape Text Detection:** Generally, arbitrary-shape scene text detection approaches can be divided into two main categories: the top-down approaches [13, 22, 23, 25, 27, 33, 40] and bottom-up approaches [10, 16, 21, 30, 31]. Top-down approaches tend to estimate the overall contour and area of the text directly or progressively. Some methods, such as those in [8, 25–27, 33], view text detection as a segmentation task, using the contours of text masks to delineate final text boundaries. Concurrently, various top-down techniques have transitioned from segmentation mask prediction to contour control point generation, employing curve functions or boundary transformation [13, 23, 37, 40]. For instance, TextRay [23] exploits Chebyshev curves, while ABCNet [13] and FCENet [40] utilize Bézier and Fourier curves, respectively. The inherent challenge for top-down methods lies in the necessity to assimilate extensive

global information and enhance the receptive field to adequately capture the final contours and areas of text.

Bottom-up methods usually begin by pinpointing potential text components and subsequently combine them, offering a counter to the challenges posed by top-down methods, notably regarding receptive field constraints and long-range dependency capture. Bottom-up methods such as [15, 21] employ heuristic rules to merge text blocks. Approaches like [16, 29, 36] make use of convolutional graph neural networks [6] to integrate text blocks. Bottom-up methods, to some extent, adopt a divide-and-conquer approach, which facilitates more flexible shape representation and higher tolerance for receptive field limitations. However, they often require complex post-processing and NMS (Non-Maximum Suppression) to generate suitable text components, leading to a rise in computational costs.

Although arbitrary-shaped text detection is a prominent research topic, current studies rarely address scenarios with insufficient illumination. Our arbitrary shape text detector adopts a bottom-up design, modeling the topological and streamline features of text under low-light conditions while optimizing the selection of text components and computational complexity.

**Low-Light Image Enhancement:** Low-light image enhancement aims to uncover image details hidden in dark areas, thereby improving image quality. Methods like KinD [38] and ZeroDCE [3] refine existing models with new training losses and heuristic quadratic curves, while approaches such as [12, 17] leverage Retinex-inspired frameworks and self-calibrated illumination for robust enhancement in low-light conditions. It is worth noting that existing low-light enhancement methods primarily aim to improve images' visual quality for human vision and often neglect the need of machine vision of the downstream tasks. Therefore, enhancing human visual experience does not necessarily translate to improved performance of the downstream machine vision tasks. In this paper, we move beyond the conventional framework of LLE-then-detection,

**Figure 2: The overall structure of the proposed method, where "1/1,256"...indicate the resize ratio and the channel number. The SCM is only employed during the training stage for assisting spatial information awareness of low-light text.**

and develop a low-light text detector based on constrained learning and effectively capturing the intrinsic characteristics of text.

**Text Detection Datasets in Low-Light Scenes:** When it comes to text detection datasets in low-light scenes, the existing datasets for arbitrary shape text detection [1, 14, 32] have been primarily collected under normal lighting conditions. Some methods such as [5, 11] have artificially introduced noise, reduce brightness, and adjust contrast to synthetically generate low-light text images based on these mainstream datasets. However, the synthetically generated low-light text images cannot fully emulate real low-light text images in terms of pixel appearances, illumination, color, and noise intensity. Additionally, synthetic methods often target on the entire image, and cannot effectively replicate the inherent visual and semantic features of low-light text.

Xue et al. [31] introduced a rudimentary dataset for low-light text detection. However, this dataset is characterized by limited diversity, including significant scene and text repetition, an average of fewer than two texts per image, and low-light text images produced by synthetically dimming daylight photos. It utilizes rectangular labeling instead of the more accurate polygonal approach and contains no curved text samples. To address these limitations and facilitate research in low-light text detection, we devise a new multilingual dataset for arbitrary shape text detection across diverse adverse scenes. Detailed description and comparison of the new dataset can be found in Section 4.

## 3 THE PROPOSED METHOD

### 3.1 Base Text Detector

The general training process of a base normal-light text detector can be formulated as:

$$\min_{\boldsymbol{\theta}} \mathcal{L}_t(\Psi(\mathbf{u}; \boldsymbol{\theta})), \tag{1}$$

where $\mathcal{L}_t$ represents the training loss used to constrain the detected output derived from the observation $\mathbf{u}$ using the text detector $\Psi$ with a learnable parameter $\boldsymbol{\theta}$. In environments degraded by low

light, the inherent process of reducing spatial dimensions in deeper feature maps of text detectors exacerbates the risk of losing or inaccurately capturing vital text spatial information, such as positional and contextual range details, thereby increasing the rate of false detections.

### 3.2 Spatial-Constrained Modeling

To address the challenges posed by low-light conditions as previously described, we design a new learning constraint that integrates spatial information into the learning process of text detector $\Psi$. This formulation can be expressed as:

$$\min_{\boldsymbol{\theta}} \mathcal{L}_t(\Psi(\mathbf{u}; \boldsymbol{\theta}(\boldsymbol{\vartheta}^*))),$$
$$s.t., \boldsymbol{\vartheta}^* = \arg\min_{\boldsymbol{\vartheta}} \mathcal{L}_s(\Phi(\mathbf{u}; \boldsymbol{\vartheta}(\boldsymbol{\theta}))). \tag{2}$$

Here, the loss $\mathcal{L}_s$ represents the newly introduced spatial constraint, designed to extract crucial position and contextual range details of text in low-light conditions from a spatial auxiliary learning module $\Phi$ (the Spatial-Constrained Learning Module (SCM) in Figure 2) with learnable parameters $\boldsymbol{\vartheta}$ to aid in the training process. The objective of the spatial constrained modeling is to identify an optimal $\boldsymbol{\vartheta}^*$ that simultaneously minimizes the loss of SCM and the base text detector. To find the optimal $\boldsymbol{\vartheta}^*$ in low-light condition, we design the spatial constraint from a dual-level perspective.

*3.2.1 Spatial Reconstruction Constraint.* The first level, termed spatial reconstruction loss $\mathcal{L}_{sr}$, is designed to enable the network to acquire a wealth of valuable information concerning the reconstruction of textual positions, thereby ensuring the preservation of textual spatial information throughout the process of reducing the dimensions of feature maps. In the training phase, we begin by creating a text position mask utilizing label information from the ground truth. Next, the output features ($C_0$ in Figure 2) are upsampled and integrated with the positional embedding via an element-wise operation. The resulting merged features are then

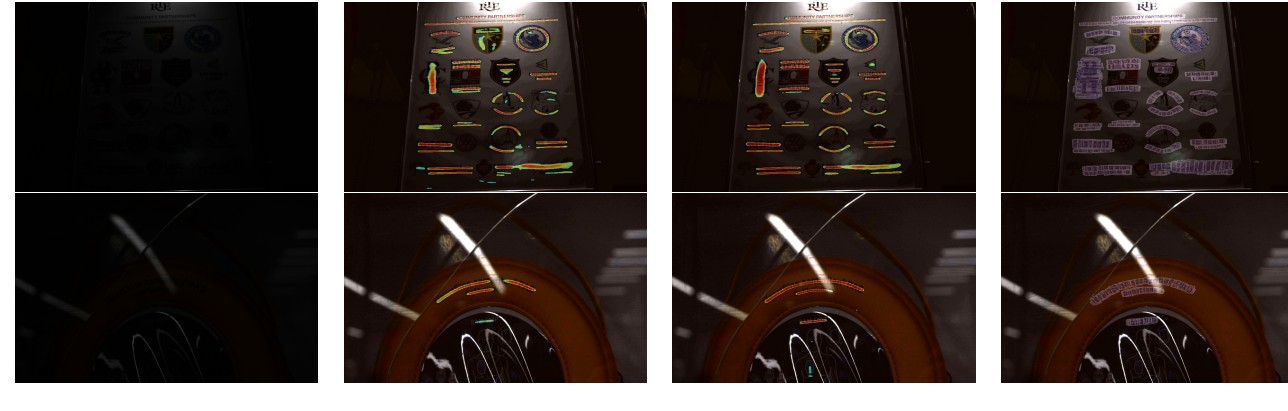

| Input | w/o SCM+DSF | SCM+DSF | TSR |

**Figure 3: Our approach, bolstered by SCM, DSF, and TSR, effectively captures both the topological structure and the center location of text components. This is demonstrated in the text center heatmaps (3rd column) and the streamlined text shaping (4th column). For clarity, images are enhanced. Here, "w/o" denotes "without".**

directed to a specially designed decoder for alignment with the ground truth.

*3.2.2 Spatial Semantic Constraint.* The second level, known as spatial semantic loss $\mathcal{L}_{ss}$, aims to align the main network and the auxiliary learning module on the contextual semantic features of text after spatial reconstruction. While the spatial reconstruction loss aids in reclaiming some lost spatial details, the semantic feature map produced by the auxiliary learning module offers essential contextual range information for text detection tasks. The $\mathcal{L}_{ss}$ can help to bridge the contextual semantic gap between the auxiliary learning branch and original text detection branch, and emphasizes the text region with greater focus in both low-light and normal-light conditions.

## 3.3 Bottom-up Text Topological Modeling

Upon introducing spatial constraints, we enhance the base text detector from the perspective of reinforcing the expression and modeling of text's topological structure, aiming to better adapt to low-light environments.

*3.3.1 Dynamic Snake FPN (DSF).* Text typically exhibits a slender distribution along its central line, with the strokes of the characters branching out in various curved directions. Therefore, we have redesigned the feature pyramid structure, leveraging the advantages of both DSC and conventional convolution by parallelizing these two convolution operations. This integration meticulously aims to capture the intricate local topological features inherent in textual elements, leveraging both the DSC branch and the regular convolution branch to enhance the detection of text's topological, visual, and semantic features.

The DSF is mainly composed of several perceptual modulation blocks stacked. As shown in Figure 2, the perceptual modulation block consists of three branches, including a DSC branch, a regular convolution branch, and a gated self-attention prediction branch, and the formulation can be written as:

$$\mathcal{X}_i = concat(C_i, \mathcal{F}_{i-1}), \qquad (3)$$

$$\mathcal{V} = concat(Conv(\mathcal{X}_i), DSC(\mathcal{X}_i)), \qquad (4)$$

$$\mathcal{F}_i = softmax\left(\frac{\sigma(W_Q \cdot \mathcal{V} + b_Q)(\sigma(W_K \cdot \mathcal{V} + b_K))^T}{\sqrt{d_k}}\right) \cdot \mathcal{V}. \quad (5)$$

Here, $\mathcal{X}_i$ represents the concatenation of feature map $C_i$ with $\mathcal{F}_{i-1}$, upon which we further concatenate the outputs from both DSC and conventional convolution applied to $\mathcal{X}_i$, denoted as $\mathcal{V}$. The $W_Q$ and $W_K$ are weight matrices for queries and keys, respectively, $b_Q$ and $b_K$ are bias terms, $\sigma$ represents the nonlinear activation function, $d_k$ are the dimensionality of the keys.

*3.3.2 Text Shaping with Rotated Rectangular Accumulation (TSR).* Following the text topology capture with DSF, we further employ a bottom-up shaping approach to enhance the expression of text's streamlined topology. Unlike some top-down methods that rely on text feature map for text shaping, the bottom-up modeling tolerates errors and needs less intact text feature maps.

For delineating text contours across various lighting conditions, our process commences with the generation of two typical text feature maps for scene text detection from the final layer of DSF: a text map and a text center region. Concurrently, this layer also generates the geometric attributes of the rotated rectangle that comprises the text component, represented by $(x, y, h, w, \theta)$. Here, $(x, y)$ designate the center of the rectangle, while $(h, w, \theta)$ determine the rectangle's height, width, and angular orientation.

Taking inspiration from the integration concept, which converts the calculation of an area into the summation of many small blocks, we accumulate the rotated rectangular text components to better fit the natural shapes of text and thereafter generate text contours. Here, we fixed the width of these rotated rectangles. While the text center region mask accurately locates the centers of the rotated rectangles, it can potentially produce an overwhelming number of candidate rectangles. Recent bottom-up approaches [16, 30, 31, 36] commonly use NMS to filter through these candidates. However, NMS does not consider the streamline distribution characteristics of text, which can result in a failure to ensure a topological distribution consistent with the central region of the text. Furthermore, NMS depends on certain degrees of randomness and parameter settings, necessitating multiple calculations of overlaps among rotated rectangles and thereby increasing the computational burden.

| Dataset | Illumination | Annotation Type | Size | | Text Length | Text Types | | | Repeat Scene/Text |
|---------|--------------|-----------------|------|------|-------------|------------|-------|-------|-------------------|
| | | | Train | Test | | Multi. | Curve | Total | |
| Dataset in [31] | Low Light | Rectangle | 300 | 200 | Short | - | - | 766 | Very High |
| CTW1500 | Normal Light | Polygon | 1,000 | 500 | Long | 7,221 | 3,530 | 10,751 | Low |
| LATeD | Low Light | Polygon | 1,000 | 500 | Long | 9,369 | 4,554 | 13,923 | Low |

Table 1: Comparison between the existing datasets and the newly constructed LATeD dataset. Multi. denotes Multi-oriented.

To address these challenges, our work abandons the NMS approach and instead adopts Farthest Point Sampling [2] to filter potential text components. Farthest Point Sampling often used in point cloud processing to reduce the size point cloud while attempting to preserve the geometrical and topological structure of the data. Here we use Farthest Point Sampling to effectively reduces the number of potential text segment centers while preserving the linear streamline characteristics of the text center regions. Here we consider $P$ as the final set of potential text components center. During each selection of potential center selection, we ensure the chosen point $p$ in $P$ satisfies the condition of being the farthest from any point in the text center region $T$. This is achieved by computing the minimum Euclidean distance $d$ to $T$:

$$p = \underset{p \in P}{\arg\max} \, \underset{t \in T}{\min} \, \|p - t\|_2. \tag{6}$$

The dotted lines (rotated rectangle center) in the 4th column of Figure 3 and 1st column in Figure 4 display the effective filtering of representative text center points using the Farthest Point Sampling technique. The cumulative shape of rotated rectangular text components accurately reflect the text's actual contours (more visual examples are in Supplementary). To accumulate these text components, our SCM approach and the DSF have also preserved the spatial position and topological structure of the text center regions. Therefore, based on this foundation, we simply need to carry out a series of straightforward morphological closing operations on each text center to fill the minor gaps between text components and within their interiors. As shown in the 3rd column of Figure 3, even when some text center regions generated with SCM and DSF support are not particularly intact due to visual degradation, the TSR still have a chance to accumulate the text regions.

### 3.4 Loss Function

The overall loss function under spatial constrains is formulated as:

$$\mathcal{L}_t = \mathcal{L}_{seg} + \mathcal{L}_H + \mathcal{L}_\theta + \mathcal{L}_{ss} + \mathcal{L}_{sr}. \tag{7}$$

Here, $\mathcal{L}_{seg}$ is the sum of cross-entropy losses for the text map and text center region. Here, $\mathcal{L}_H$ and $\mathcal{L}_\theta$ represent the smoothed $L1$ losses for the height and rotation angle of the text component, respectively. The $\mathcal{L}_{ss}$ is $L2$ loss and the $\mathcal{L}_{sr}$ is the $L1$ loss for the text spatial constraint. The first three losses in the $\mathcal{L}_t$ are used to supervise the computation of rotated rectangular text components, while the last two losses represent the spatial constraints we introduced.

## 4 LATED: LOW-LIGHT ARBITRARY-SHAPE TEXT DETECTION DATASET

The existing low-light text detection dataset [31] presents several issues: on average, it contains fewer than two texts per image, exhibits significant scene and text repetition, uses rectangular labeling

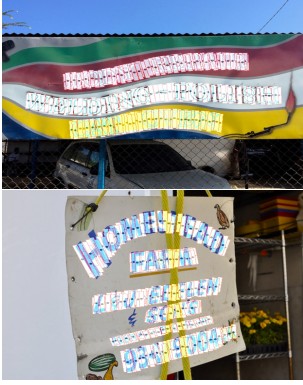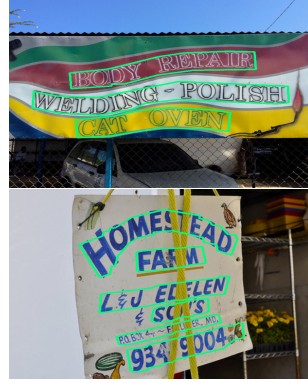

Figure 4: Text shaping with rotated rectangular accumulation on normal light text images. As shown in the 2nd row, our method remains unaffected by obstructions, showing reliable performance in modeling text streamline features.

instead of the more precise polygon labeling, and is limited in size with no curved text samples. Additionally, ambiguity regarding the composition of the test and training sets, along with the absence of specified evaluation criteria, hinders the effective use and reproducibility of this dataset. These challenges significantly impede the development of low-light arbitrary scene text detection.

Therefore, we created LATeD, a new dataset manually curated during nighttime from multilingual communities in Sydney and Melbourne, and tailored specially for arbitrary-shape scene text detection in low-light environments.

Drawing inspiration from the flagship dataset CTW1500 [14], which was created for text detection under normal lighting conditions, LATeD surpasses CTW1500 in both the total text count and the number of curved texts (see Table 1). The LATeD dataset consists of 1,500 images (1,000 for training and 500 for testing), featuring 13,923 arbitrarily shaped texts, with 4,554 being curved texts. Each text is accompanied by precise line-level polygon annotations, providing a valuable resource for low-light text detection research.

Furthermore, LATeD is a multilingual dataset, primarily composed of English and Chinese texts while also containing languages such as Japanese, Korean, Vietnamese, and Arabic. The dataset covers a broad spectrum of low-light scenes, ranging from indoor to outdoor settings and encompassing various mediums such as standard printed posters to packages, clothes, billboards, road signs, graffiti, and more.

Detailed statistics regarding light condition, scene and languages of the two datasets can be found in Table 1 with the visual statistics and examples of images in the LATeD dataset shown in Figure 1. Notably, every image in LATeD was 100% manually curated during

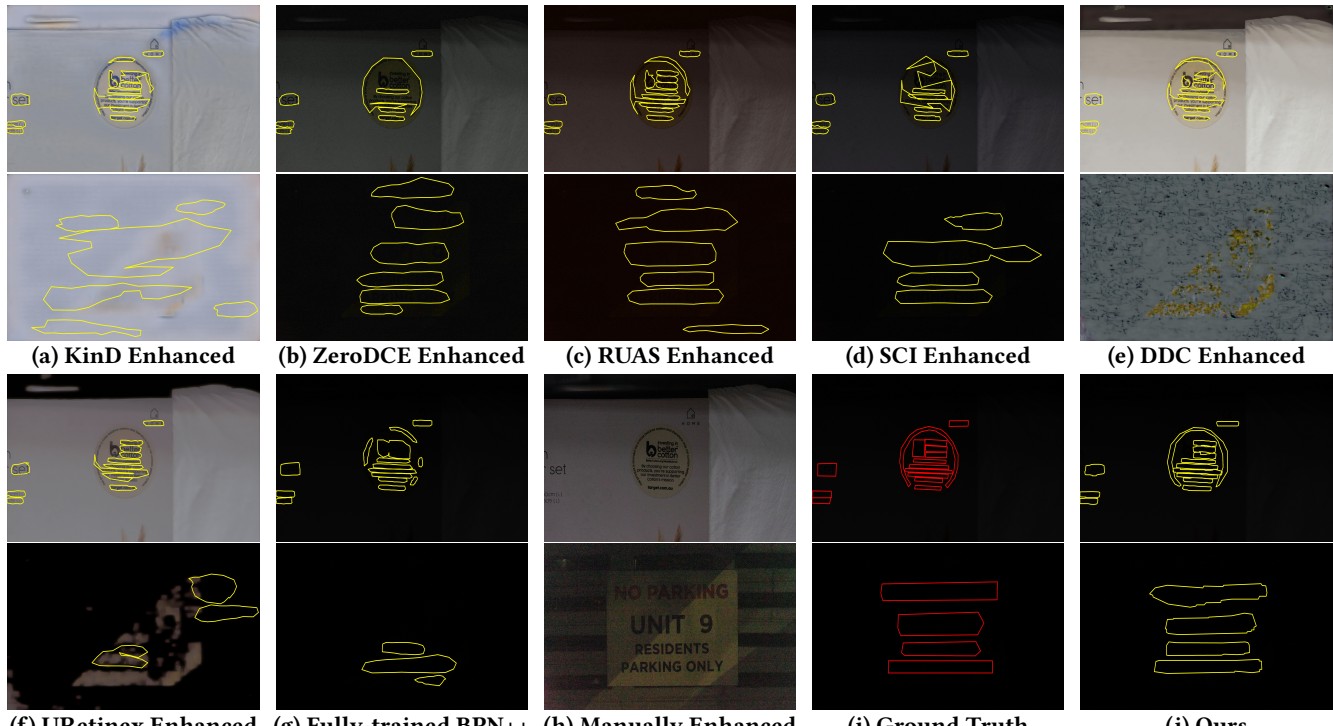

**(a) KinD Enhanced** **(b) ZeroDCE Enhanced** **(c) RUAS Enhanced** **(d) SCI Enhanced** **(e) DDC Enhanced**

**(f) URetinex Enhanced** **(g) Fully-trained BPN++** **(h) Manually Enhanced** **(i) Ground Truth** **(j) Ours**

**Figure 5: Visual comparison of the text detection results on images from the LATeD dataset. (a)-(f) Texts detected with BPN++ fine-tuned using images enhanced with six different LLE techniques. (g) Texts detected with fully trained BPN++ using low-light images. (h) Manually enhanced images. (i) Input low-light images with ground-truth bounding boxes. (j) Text detected with our method.**

nighttime from multilingual communities in Sydney and Melbourne. To assess text detection accuracy, we simply follow the evaluation protocol from CTW1500.

## 5 EXPERIMENTS

To demonstrate the superior performance of the proposed text detector, we conducted series of experiments comparing our approach with SOTA methods. For low-light text detection, we utilized the newly created low-light text dataset LATeD. For normal-light text detection, we utilized CTW1500, Total-text [1] and MSRA-TD500 [32].

### 5.1 Implementation Details

The backbone of our network is ResNet50. We first pre-trained it on the SynthText dataset [4] for 2 epochs using input images of size 640 × 640 pixels. Subsequently, we fine-tuned it for an additional 100 epochs on the MLT dataset [18]. We employed the Adam optimizer with a learning rate initialized at 0.0001, which was reduced by a factor of 0.1 every 100 epochs. Our data augmentation strategy included random cropping, resizing, color adjustments, noise injection, flipping, and rotation. The batch size was set to 10 and the training epoch was set to 250. The training was conducted on a single NVIDIA RTX A6000 GPU, supported by a 3.60GHz Intel Xeon Gold 5122 CPU. During testing, input images were resized to 640 × 640 for LATeD and CTW1500, and 640 × 1200 for Total-Text and 640 × 960 for MSRA-TD500.

## 5.2 Comparison on Low-Light Text Detection

We benchmarked eleven state-of-the-art (SOTA) methods for arbitrary shape scene text detection and six SOTA low-light enhancement techniques across four settings: text detection on low-light images ("Detection"), text detection on enhanced images ("LLE"), text detector fine-tuned with enhanced images ("FINE-TUNE"), and fully trained text detector ("FULLY-TRAINED"). Table 2 compares the detailed detection results of our approach with the SOTA approaches under all four settings. Figure 5 visualize the text detection results of two exemplar images from the LATeD dataset using different approaches.

*Setting 1: Text Detection on Low-light Images (Detection):* we first tested eleven text detectors originally designed for normal light conditions (pretrained on the CTW1500) on low-light images. As shown in Table 2, unsurprisingly, all of the examined text detectors performed poorly under low-light conditions with none exceeding 50% F1 scores and BPN++ performing the best. This shows the limited generalizability of existing text detectors in dealing with low-light conditions.

*Setting 2: Text Detection on Enhanced Images (LLE):* We then applied six different LLE methods to enhance the images in the LATeD dataset, and used BPN++ as the baseline text detector due to its superior F1-score, indicating better robustness and generalizability. As shown in Figure 5 and Table 2 (the "LLE" section), although most enhancement techniques managed to improve the image visibility for human eyes, the improvement is insufficient for the downstream

| Task | Method | Venue | LATeD | | |
|---|---|---|---|---|---|
| | | | P(%) | R(%) | F1(%) |
| *Scene text detection* (Detection) | | | | | |
| Detection | PSENet [25] | CVPR'19 | 53.7 | 7.5 | 13.1 |
| | PAN [26] | ICCV'19 | 56.4 | 8.1 | 14.2 |
| | DB [7] | AAAI'20 | 4.6 | 9.8 | 13.2 |
| | DRRG [36] | CVPR'20 | 73.2 | 12.2 | 20.9 |
| | ContourNet [36] | CVPR'20 | 61.2 | 19.5 | 24.1 |
| | TextFuseNet [33] | IJCAI'20 | 21.8 | 91.8 | 37.4 |
| | FCEnet [40] | CVPR'21 | 15.5 | 72.4 | 32.1 |
| | TextBPN [37] | ICCV'21 | 21.9 | 65.9 | 42.7 |
| | DB++ [8] | TPAMI'22 | 61.2 | 19.6 | 29.6 |
| | DPText [34] | AAAI'23 | 86.7 | 30.6 | 45.3 |
| | BPN++ [35] | TMM'23 | 73.6 | 35.3 | **47.7** |
| *Text detection on enhanced images* (LLE) | | | | | |
| LLE | KinD [38] | MM'19 | 88.4 | 35.9 | 51.1 |
| | ZeroDCE [3] | CVPR'20 | 90.8 | 37.7 | 53.3 |
| | RUAS [12] | CVPR'21 | 90.4 | 36.5 | 52.0 |
| | SCI [17] | CVPR'22 | 91.1 | 34.0 | 49.5 |
| | DDC [39] | CVPR'22 | 88.6 | 39.1 | 54.2 |
| | URetinex [28] | CVPR'22 | 89.4 | 34.3 | 49.6 |
| *Fine-tuned on enhanced images* (FINE-TUNE) | | | | | |
| Fine-tune | KinD [38] | MM'19 | 76.0 | 43.6 | 55.4 |
| | ZeroDCE [3] | CVPR'20 | 78.1 | 46.6 | 59.1 |
| | RUAS [12] | CVPR'21 | 79.1 | 48.1 | 59.8 |
| | SCI [17] | CVPR'22 | 78.0 | 47.7 | 59.2 |
| | DDC [39] | CVPR'22 | 75.2 | 47.5 | 58.2 |
| | URetinex [28] | CVPR'22 | 72.5 | 40.8 | 52.4 |
| *Fully trained text detectors* (FULLY-TRAINED) | | | | | |
| Fully Trained | BPN++ [35] | TMM'23 | 74.5 | 49.5 | 59.5 |
| | **Ours** | - | **82.6** | **57.0** | **67.1** |

**Table 2: Quantitative comparison of text detection results obtained on the low-light text detection dataset LATeD under the four settings.**

detectors to perceive the text clearly. These methods fall short when compared to ours, as they introduced too many visual distortions and still exhibit gaps with normal light images.

*Setting 3: Fine-tuned Text Detectors with Enhanced Images (FINE-TUNE):* To further validate the effectiveness of the "enhance-first, detect-later" approaches, we enhanced the training images of the LATeD dataset using various LLE methods [3, 12, 17, 28, 38, 39] and then fine-tuned the BPN++ on the enhanced training images.

As shown in Figure 5 (a)-(f), some LLE methods such as DDC, KinD, and URetinex introduced visual distortions, impairing images' textual information and leading to direct text detection failures. Other methods such as RUAS and SCI enhanced image brightness, but misled the downstream text detector at text boundary areas. These results, however, still cannot match the performance of our method, which effectively detects text without requiring any LLE module. The primary reason is that those enhancement techniques are not designed with downstream tasks in mind, leading to a semantic gap between enhanced low-light images and those taken in normal lighting conditions.

*Setting 4: Fully Trained Text Detector (FULLY-TRAINED):* Given that LLE techniques are not tailored for downstream text detection, we trained BPN++ and our method directly on the low-light images of the LATeD dataset. As shown in Table 2, the F1-scores obtained with the LLE methods are close to that of the BPN++, suggesting LLE is tuned for enhanced human visual perception rather than maximizing downstream performance. Moreover, general text detectors such as BPN++ lack targeted guidance on text spatial information during training for low-light settings, hence falling short in effectively expressing the local topology and streamline features of text. This results in a performance that was not on par with our purposefully designed method. The detection results shown in Figure 5 also demonstrate the robustness of our method for detecting curved texts in low-light environments.

## 5.3 Comparison on Well-lit Text Detection

In this section, we present experimental results for text detection under normal lighting using the CTW1500, Total-Text [1], and MSRA-TD500 [32] datasets, following their official evaluation protocols. Table 3 presents the details results, where "Extend" indicates the additional datasets used for pre-training in comparative methods. In our implementation, we used ResNet50 as the primary backbone for all networks, except for those mentioned in [7] and [8], which used ResNet50 with deformable convolution.

According to Table 3, our method achieved an F1-score of 86.2% on CTW1500, surpassing current state-of-the-art methods such as [20, 35]. Additionally, for arbitrarily shaped short texts at the word level, our approach established a new state-of-the-art on the Total-Text dataset with an F1-score of 88.5% and the highest recall of 86.6%. For long multi-oriented texts under normal lighting, our method achieved a performance comparable to the state-of-the-art on the MSRA-TD500 dataset.

The effectiveness of our method is attributed to the combined use of SCM, DSF, and TSR, crucial for capturing text's topological and streamline features in normal lighting. This demonstrates our method's potential to deliver high performance and efficiency in both normal light and low light conditions. Examples of visual detection results can be found in Figure 6 and supplementary.

## 5.4 Ablation Study

We further conducted a series of ablation studies on LATeD and CTW1500 to validate the effectiveness of the proposed modules, TSR, DSF and SCM, comparing them with a baseline with a standard FPN and 0.5 threshold NMS for rotated rectangle text components but excluding the SCM.

As shown in Table 4, adding the text shaping module TSR into the baseline model yielded a 0.8% and 0.3% increase in F1-score on LATeD and CTW1500, respectively, and greatly enhanced inference speed. This improvement can be attributed to the farthest points sampling, which helps maintain the text component's topological distribution. Our method retained a set number of text components, avoiding the extensive overlap calculations for numerous candidates required by NMS, thus significantly reducing computation.

Furthermore, after implementing the SCM and TSR, the model's F1-score improved by 4.4% and 1.9%, respectively, without any reduction in inference speed. SCM, during training, provides cues

| Method | Venue | Extend | CTW1500 | | | Total-Text | | | MSRA-TD500 | | |
|---|---|---|---|---|---|---|---|---|---|---|---|
| | | | P(%) | R(%) | F1(%) | P(%) | R(%) | F1(%) | P(%) | R(%) | F1(%) |
| PSENet [25] | CVPR'19 | MLT | 86.9 | 80.2 | 83.4 | 84.0 | 78.0 | 80.9 | - | - | - |
| DB [7] | AAAI'20 | SynthText | 86.9 | 80.2 | 83.4 | 87.1 | 82.5 | 84.7 | 91.5 | 79.2 | 84.9 |
| ContourNet [27] | CVPR'20 | - | 83.7 | 84.1 | 83.9 | 86.9 | 83.9 | 85.4 | - | - | - |
| TextFuseNet[33] | IJCAI'20 | SynthText | 85.0 | 85.8 | 85.4 | 87.5 | 83.2 | 85.3 | - | - | - |
| DRRG [36] | CVPR'20 | MLT | 85.9 | 83.0 | 84.4 | 86.5 | 84.9 | 85.7 | 88.1 | 82.3 | 85.1 |
| FCENet [40] | CVPR'21 | - | 85.7 | 80.7 | 83.1 | 87.4 | 79.8 | 83.4 | - | - | - |
| TextBPN [37] | ICCV'21 | MLT | 86.5 | 83.6 | 85.5 | 90.7 | 85.2 | 87.9 | 86.6 | 84.5 | 85.6 |
| FSGNet [22] | CVPR'22 | MLT | 88.1 | 82.4 | 85.2 | 90.7 | 85.7 | 88.1 | - | - | - |
| DB++ [8] | TPAMI'22 | SynthText | 87.9 | 82.8 | 85.3 | 88.9 | 83.2 | 86.0 | 91.5 | 83.3 | 87.2 |
| SIR [20] | MM'23 | Syn | 87.4 | 83.7 | 85.5 | 90.9 | 85.6 | 88.2 | 93.6 | 86.0 | 89.6 |
| MTM [24] | MM'23 | Syn | 85.8 | 83.4 | 84.6 | 89.6 | 82.1 | 85.7 | 90.3 | 81.4 | 85.6 |
| BPN++ [35] | TMM'23 | MLT | 87.3 | 83.8 | 85.5 | 91.8 | 85.3 | 88.5 | 89.2 | 85.4 | 87.3 |
| **Ours** | - | MLT | 88.7 | 83.9 | **86.2** | 90.4 | **86.6** | **88.5** | 91.0 | 83.5 | 87.1 |

**Table 3: Text detection results on CTW1500, TOTAL-TEXT and MSRA-TD500.**

| SCM | DSF | TSR | LATeD | | | | CTW1500 | | | |
|---|---|---|---|---|---|---|---|---|---|---|
| | | | FPS | P(%) | R(%) | F1(%) | FPS | P(%) | R(%) | F1(%) |
| × | × | × | 2.1 | 78.8 | 47.9 | 59.6 | 2.2 | 84.1 | 80.6 | 82.3 |
| × | × | ✓ | 10.9 | 80.8 | 48.2 | 60.4 | 11.1 | 84.2 | 81.0 | 82.6 |
| ✓ | × | ✓ | 10.9 | 81.5 | 52.7 | 64.0 | 11.1 | 85.1 | 83.4 | 84.2 |
| × | ✓ | ✓ | 10.2 | 79.8 | 51.1 | 62.3 | 10.4 | 84.9 | 82.9 | 83.8 |
| ✓ | ✓ | ✓ | 10.2 | 82.8 | 55.4 | 66.4 | 10.4 | 86.2 | 83.5 | 84.8 |

**Table 4: Ablation study on the effectiveness of the proposed SCM, DSF and TSR on the LATeD and CTW1500 dataset.**

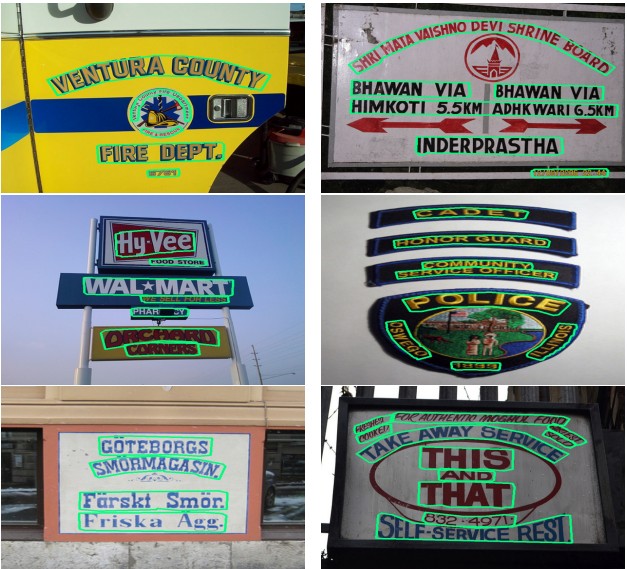

**Figure 6: Visualization of text detection results on well-lit datasets (see Supplementary for more visual examples).**

about the text's spatial information, aiding the network in perceiving text locations under low contrast and degraded low light conditions. Similarly, under normal lighting, SCM continues to provide spatial information for the text detection task, facilitating precise detection of the center location of text components.

After replacing the FPN structure with our designed DSF and incorporating our text shaping method, the F1-score improved by 2.7% and 1.5% on the two datasets, respectively. These improvements are due to the DSF's adaptive weighting of regular convolution and DSC operations, focusing on capturing the topological features of text, thereby generating more reliable text center region and geometric features of text components. As demonstrated in the third column of Figure 3, the absence of SCM and DSF led to ineffective detection of the topological structure of text center in low-light conditions, resulting in low-confidence outcomes and inaccurate disconnections of text center.

Utilizing SCM, DSF, and TSR, our model achieved F1-scores of 66.4% and 84.8% on the two datasets, respectively. This represents an increase of 6.8% and 2.5% in F1-score over the baseline. The efficacy of our method in both low light and normal light conditions stemmed from its precision in providing spatial cues for text location and its capacity to preserve text's topological streamline and features. When allied with a bottom-up design approach, these attributes were instrumental in sustaining long-range dependencies, crucial for precise text detection. The fourth column's results highlighted in Figure 3 and second row's results in Figure 4 show the TSR method's effectiveness in sampling text components and maintaining text streamline features.

## 6 CONCLUSION

In this work, we crafted a constrained learning module that capitalizes on spacial information of text, thereby enhancing the efficacy of text detectors in low-light environments. Our approach, integrating Dynamic Snake FPN with a rectangular, bottom-up text contour shaping method, marks a significant advancement in accurately representing text's topological distribution and streamline features. This innovative methodology has enabled us to achieve state-of-the-art results in both low-light and normal-light text detection datasets. In addition, we have created an extensive and diverse dataset for arbitrary-shaped text specific to low-light conditions, encompassing a broad spectrum of scenes and languages, thereby significantly enhancing the resources available in this research area.

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
