# OpenReview forum: "Seeing Text in the Dark: Algorithm and Benchmark"
_acmmm.org/ACMMM/2024/Conference — MM2024 Poster_

### Official Review · Reviewer_TB4y · 2024-05-07

**Rating:** 2
**Confidence:** 3

**Summary:**

This paper proposes a new dataset tailored for low-light arbitrary-shape text detection called LATeD. This is a multilingual dataset containing images in low-light conditions from many countries in many language. Aside from it, this paper also propose a model to solve the problem of low-light arbitrary-shape text detection. It uses the spatial-constrained modeling strategy and bottom-up text topological modeling method. Experimental results show that this model achieves superior results in this task.

**Strengths:**

1. The proposed dataset seems to contain sufficient images in various conditions to facilitate the task of low-light arbitrary-shape text detection.
2. The proposed model jumps out of the box of the paradigm of normal text detection methods and introduces a spatial-constrained method.
3. The proposed model achieves superior results on many tasks and datasets.

**Limitations:**

1. I have some questions with the setting of this work. The title of this paper is "Seeing Text in the Dark", from my point of view, "seeing" including "recognition" and "detection". However, this paper only focuses on low-light arbitrary-shape text detection. I wonder why the authors only focus on detection instead of recognition or spotting? For low-light text recognition, there has been some previous works, such as Nguyen et al. [1]. I think the authors should thoroughly explain their setting because I think the setting now is not complete at all.

2. The authors provide some information on the constructed dataset on Table 1. However, I think more information are obligated to be provided. For example, the authors mentioned that this dataset contains images with many languages in L571. Therefore, I think the authors should display how many images there are for different languages. Also, the authors only mention that the dataset contains images from various scenes in L571-575. To me, the authors should provide detailed statistics on how many images there are for indoor and outdoor settings, and how many images there are in mediums such as standard printed posters to packages, clothes, billboards, road signs, graffiti, and others. Additionally, the authors should display how they collect these images, whether they are collected from other datasets, or shot by the authors via which camera or mobile phone? And how they construct the labels? These key information are all missing in the paper.

3. For the proposed model, the authors state that the SCM is capable of assisting spatial information awareness of low-light text and a spatial reconstruction loss is introduced. Therefore, I want to see the accuracy of between the predicted result of SCM and the ground truth.

4. In L473-480, the authors state that the Farthest Point Sampling method is better that NMS. However, no experimental result is provided to support this point. Therefore, I think ablation studies are required to be conducted to illustrate that Farthest Point Sampling method is truely better that NMS.

5. The authors use Dynamic Snake Convolution (DSC) in this paper. I wonder why the authors use this kind of convolution? Did they employ other kinds of convolutional layers for testing? I think detailed ablation study should be provided.

6. In Eq.(7), the authors fix the weights of every loss function as 1 and I do not think it is rigorous at all. To me, Eq.(7) should be changed to $\mathcal{L}\_{\mathrm{t}}=\lambda\_1\mathcal{L}\_{\mathrm{seg}}+\lambda\_2\mathcal{L}\_{H}+\lambda\_3\mathcal{L}\_{\theta}+\lambda\_4\mathcal{L}\_{\mathrm{ss}}+\lambda\_5\mathcal{L}\_{\mathrm{sr}}$ and the authors should provide experimental results to find the best values of $\lambda\_1$ to $\lambda\_5$.

7. The writing quality of this paper is somewhat poor. I think more polish is required before this paper is qualified for publication:

   - In L250, the style of quotation mark before and after "1/1,256" is different.
   - In L429, the latex code should be "C_i" and "F_{i-1}".
   - Some languages in this paper is informal. For example, in L788, the authors use "text’s topological and streamline features". This abbreviation is very informal in such a technical paper and it should be replaced by "the topological and streamline features of text". This problem exists throughout the whole paper and the authors should take time to check it.
   - I recommend the authors to rewrite the third section because I think this part is difficult to read. For example, all the figures in this paper are not vectorgraph, which is not good. The authors also mention in L400 that DSF has three branches, however, none of them is labeled in Figure 3, which makes me difficult to understand. The authors also mention "$\mathcal{C}\_0$" in L345, however, details of the model are illustrated in the following parts. Therefore, I think the authors should regroup the third section to make it easier to read.

   **References:**

   [1] Nguyen et al. Diffusion in the Dark: A Diffusion Model for Low-Light Text Recognition. WACV 2024.

**Suitability:**

2

---

### Official Review · Reviewer_Sik4 · 2024-05-16

**Rating:** 4
**Confidence:** 4

**Summary:**

This paper proposed a one-stage pipeline for low-light text detection. It used spatial constraint to guide the training process. It concentrated on the extraction of topological distribution features and the modeling of streamline characteristics to shape low-light text contours. A low-light arbitrary-shaped text dataset is proposed, which contains 13923 texts across diverse scenes. The proposed method achieves competing results on the low-light text detection dataset.

**Strengths:**

Ablation Study: The paper includes a thorough ablation study, as shown in Table 4, which substantiates the contribution of the proposed SCM, DSF, and TSR. This study is crucial as it isolates the impact of each component, allowing for a clear understanding of their individual and collective contributions to the overall performance.


State-of-the-Art Results: The method has achieved state-of-the-art (SOTA) results on the low-light text detection dataset, which is a strong testament to its effectiveness. Achieving SOTA results is a significant milestone as it positions the method at the forefront of current research in this domain.


New Dataset: The introduction of a new dataset for low-light arbitrary-shaped text detection is a substantial contribution to the research community. This dataset not only provides a benchmark for evaluating new methods but also encourages further development and refinement of existing techniques.

**Limitations:**

Discrepancy in Results: There appears to be a discrepancy between the results presented in Table 3 and Table 4, specifically the performance on the CTW1500 dataset, where the reported percentages are 86.2% and 84.8%, respectively. It would be beneficial to understand the differences between the models or conditions that led to these variations. Clarifying this discrepancy would strengthen the reliability of the reported results.


Annotation Quality Assurance: The paper raises a valid concern regarding the quality of annotations in the proposed dataset. Given the low-light conditions, it can be challenging for human annotators to accurately read and label the text. The authors should address how they ensured the quality and accuracy of the annotations, possibly through multiple rounds of verification or the use of advanced tools to assist in the annotation process.

**Suitability:**

2

---

### Official Review · Reviewer_CJbS · 2024-05-28

**Rating:** 4
**Confidence:** 3

**Summary:**

This paper focuses on text detection in low-light scenes. First, they create a new low-light text detection dataset for training. Then, they propose a CSM module to align the feature extracted from the input image and label. As a result, the proposed model outperforms other existing text detectors on both low-light dataset and normal datasets.

**Strengths:**

1. The proposed dataset fills the blank of low-light text detection.
2. The proposed model gets SOTA performance on both low-light dataset and some normal datasets.
3. The paper is well-organized and easy to follow.

**Limitations:**

1. It seems that the prediction for Spatial Constraint uses the feature extracted from the label (In the “element wise addition” of Fig.2). How to ensure this constraint loss can optimize the feature F1 in the detection model.
2. The key module in DSF is the existing Dynamic Snake Convolution method.
3. What is the training set of Table.4? It seems that the proposed dataset only can bring 0.7 improvement in low-light text detection (66.4 to 67.1)
4. Table.1 lacks the comparison of the baseline model in all 4 settings.

**Suitability:**

2

---

### Meta-Review · Area_Chair_DaSt · 2024-07-03

**Recommendation:** Accept (Poster)
**Confidence:** 4

**Metareview:**

This paper focuses on text detection in low-light scenes, by proposing a new training dataset and a text detector. According to the 3 reviewers, the paper is well-written, clearly organized and reaches SOTA performance on both low-light dataset and some other datasets. Most of the main questions raised are addressed by the rebuttal, even if the justification for the absence of some experiments (additional ablation studies) cannot be justified by the length imposed, given the possibility of depositing supplementary material. It seems to me that this work deserves to be published at ACM MM, perhaps as a poster, because of the question of its relevance to the community, since it is a single-modality work, for very targeted data and applications.